# Lactic Acid Resistance and Population Structure of *Escherichia coli* from Meat Processing Environment

Yuan Fang,[a] Kim Stanford,[b] Xianqin Yang[a]

[a]Agriculture and Agri-Food Canada Lacombe Research and Development Centre, Lacombe, Alberta, Canada
[b]University of Lethbridge, Lethbridge, Alberta, Canada

**ABSTRACT** To explore the effect of beef processing on *Escherichia coli* populations in relation to lactic acid resistance, this study investigated the links among acid response, phylogenetic structure, genome diversity, and genotypes associated with acid resistance of meat plant *E. coli*. Generic *E. coli* isolates ($n = 700$) were from carcasses, fabrication equipment, and beef products. Acid treatment was carried out in Luria-Bertani broth containing 5.5% lactic acid (pH 2.9). Log reductions of *E. coli* ranged from <0.5 to >5 log CFU/mL (median: 1.37 log). No difference in lactic acid resistance was observed between *E. coli* populations recovered before and after a processing step or antimicrobial interventions. *E. coli* from the preintervention carcasses were slightly more resistant than *E. coli* isolated from equipment, differing by <0.5 log unit. Acid-resistant *E. coli* (log reduction <1, $n = 45$) had a higher prevalence of genes related to energy metabolism (*ydj, xap, ato*) and oxidative stress (*fec, ymjC*) than the less resistant *E. coli* (log reduction >1, $n = 133$). The *ydj* and *ato* operons were abundant in *E. coli* from preintervention carcasses. In contrast, *fec* genes were abundant in *E. coli* from equipment surfaces. The preintervention *E. coli* contained phylogroups A and B1 in relatively equal proportions. Phylogroup B1 predominated (95%) in the population from equipment. Of note, *E. coli* collected after sanitation shared either the antigens of O8 or H21. Additionally, genome diversity decreased after chilling and equipment sanitation. Overall, beef processing did not select for *E. coli* resistant to lactic acid but shaped the population structure.

**IMPORTANCE** Antimicrobial interventions have significantly reduced the microbial loads on carcasses/meat products; however, the wide use of chemical and physical biocides has raised concerns over their potential for selecting resistant populations in the beef processing environment. Phenotyping of acid resistance and whole-genome analysis described in this study demonstrated beef processing practices led to differences in acid resistance, genotype, and population structure between carcass- and equipment-associated *E. coli* but did not select for the acid-resistant population. Results indicate that genes coding for the metabolism of long-chain sugar acids (*ydj*) and short-chain fatty acids (*ato*) were more prevalent in carcass-associated than equipment-associated *E. coli*. These results suggest *E. coli* from carcasses and equipment surfaces have been exposed to different selective pressures. The findings improve our understanding of the microbial ecology of *E. coli* in food processing environments and in general.

**KEYWORDS** *E. coli*, acid resistance, beef safety, population structure

Address correspondence to Xianqin Yang, xianqin.yang@agr.gc.ca.

The authors declare no conflict of interest.

*E*scherichia coli is an inhabitant of the intestines of warm-blooded animals (1), but some strains are pathogenic to humans and animals (2). Cattle are well-recognized hosts of Shiga toxin-producing *E. coli* (STEC), including *E. coli* O157:H7, making beef products susceptible to STEC contamination (3, 4). Fecal shedding of *E. coli* results in a heavy microbial load on animal hides, particularly animals raised in centralized feedlot environments (5 to 7). The transmission of *E. coli* to beef carcasses is regarded as

inevitable during the process of dehiding because the opening cuts are from the outer to the inner surface of the hides and aerosols are generated by the motion of hide pulling (7, 8). Concerns over STEC and other microbiological hazards led to the requirement in the United States for testing of carcasses for *E. coli*, and beef trimmings for *E. coli* O157:H7 and six non-O157 STEC (9). The prevalence and level of pathogenic *E. coli* on carcasses are much lower than generic *E. coli*, which could render population-based studies impossible without sampling an extremely high number of samples. Generic *E. coli* is thus an appropriate indicator to assess the hygiene performance of meat processing (10).

To address concerns regarding the microbiological safety of meat, various physical and chemical antimicrobial interventions are implemented at beef plants in North America (7, 11, 12). Commonly used antimicrobial interventions in large beef processing facilities include hide-on carcass wash with 1.5% sodium hydroxide at 55°C (13, 14), carcass wash with organics acids, such as 4 to 5% lactic acid and peroxyacetic acid at various concentrations, and pasteurization of carcasses with steam or hot water at 85°C or >90°C (15). Air chilling can effectively reduce the microbial load on carcasses by up to 2 log units (16) and can also be regarded as an intervention step. *E. coli* on most chilled carcasses can be reduced to <1 CFU/10,000 cm$^2$ when the microbial contamination of meat during carcass dressing is well controlled and decontamination for carcasses is effective (17). Consequently, the decreased number of foodborne illnesses caused by *E. coli* O157:H7 in Canada has been attributed to the implementation of antimicrobial interventions at beef processing plants (18).

Lactic acid at 4 to 5% is one of the most effective chemical interventions for reducing the microbial load of carcasses, including *E. coli* in commercial beef processing settings (7, 17). However, complete elimination of bacteria on carcasses by any antimicrobial interventions is not yet attainable, largely due to the structure and/or composition of carcass surfaces and raw meat (13, 17). In addition to the application of antimicrobial interventions to decontaminate carcasses/meat products, various biocides, such as caustic chlorine cleaners, quaternary compounds, and peroxyacetic acid-based sanitizers are routinely used to clean and sanitize equipment to minimize recontamination of meat during meat fabrication (12, 19). Accordingly, the wide use of physical/chemical agents in meat processing facilities may select for bacteria with elevated resistance to antimicrobial interventions.

*E. coli* is generally acid tolerant (20 to 22). To survive in acidic environments, *E. coli* utilizes various physiological mechanisms coupled with the buffering capacity of cytoplasmic macromolecules, such as proteins, amino acids, and polyphosphate, and the reduction of proton influx mediated by membrane fluidity and cytoplasmic chaperones, as well as consumption of intracellular protons through decarboxylases systems (23). Additionally, secondary carbohydrate metabolism and ATP-dependent metabolic pathways have also been suggested to be involved in supporting *E. coli* survival under acidic conditions (24). The ability to balance intracellular pH and to recover from the cellular damages caused by low pH enhances the survival of *E. coli* in acid conditions. If the current meat packing practices with respect to antimicrobial interventions and usage of biocides in sanitation lead to increased acid resistance in *E. coli* in the beef processing environment, it would not only compromise the efficacy of lactic acid as an antimicrobial intervention but may also contribute to the increased prevalence of acid-resistant *E. coli* in the broader environment. Therefore, the present study aimed to (i) determine the acid response of 700 *E. coli* collected from different stages of beef processing to evaluate whether the beef processing environment selects for acid-resistant *E. coli*, and (ii) explore the relationships among the acid phenotypes, phylogeny, source of isolation, and genetic makeup of *E. coli* to unravel the mechanisms driving differences in acid resistance.

## RESULTS

**Relative response of *E. coli* to lactic acid treatment.** To determine whether processing stages in beef slaughter select for acid-resistant *E. coli*, the cell-count reduction of *E. coli* by the lactic acid treatment (pH 2.9 for 1h) was determined and compared

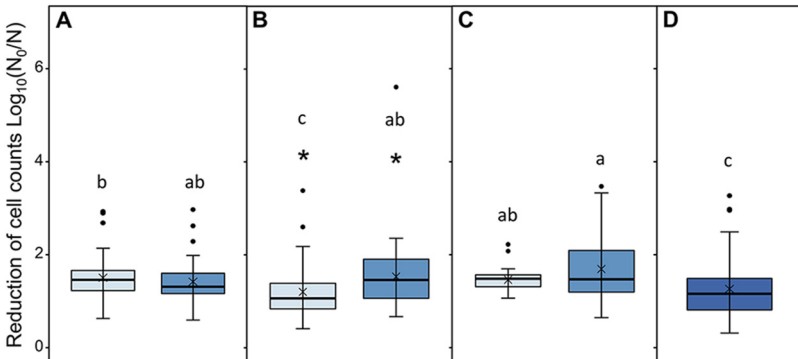

**FIG 1** Lethality (log$_{10}$ N$_0$/N) of lactic acid treatment to *E. coli* recovered from different stages of beef processing. Panels show *E. coli* recovered from carcasses before and after hide-on wash (A), carcasses before and after air chilling (B), equipment surfaces before and after sanitation (C), and meat products (D). In panels A to C, light-blue and dark-blue were used to differentiate before and after a process, respectively. The median log reductions are indicated by a line; the mean log reductions are indicated by ×; whiskers indicate data within 1.5 times the interquartile range (IQR) above the first quartile and 1.5 times IQR below the third quartile; and outliers are indicated by the filled dots. Each population included 100 *E. coli* isolates, and two independent experiments, each including two technical replicates, were performed for each isolate. A single asterisk indicates significant differences in the reduction of cell counts of the *E. coli* populations from before and after a process (panels A to C); different letters indicate significant differences among the seven *E. coli* populations (panels A to D; $P < 0.05$).

between and among populations from different processing stages (Fig. 1). The log reductions of *E. coli* resulting from acid treatment ranged from <0.5 to >5 log$_{10}$ CFU/mL, with the median being 1.37 log. Median log reductions for the *E. coli* populations from carcasses before and after hide-on carcass wash (1.35 and 1.55 log unit) or from equipment surfaces before and after equipment sanitation (1.42 and 1.48 log unit) did not differ significantly ($P > 0.05$). Isolates from air-chilled carcasses had an increased sensitivity to the lactic acid treatment ($P < 0.05$) as the median value of log reductions increased from 1.07 to 1.49 log unit. For the *E. coli* populations from different stages of processing, *E. coli* recovered from carcass surfaces before air chilling and meat cuts/trim had slightly lower ($P < 0.05$) reductions than the other *E. coli* populations, by up to 0.5 log unit.

To determine the effect of the source of isolation and antimicrobial interventions on the susceptibility of *E. coli* to the lactic acid treatment, data were reanalyzed to compare the *E. coli* populations associated with animals, i.e., preintervention carcasses and equipment surfaces, or the *E. coli* populations from carcasses treated and not treated with antimicrobial interventions (Fig. S1). The preintervention carcass-associated *E. coli* population had a slightly but statistically significant ($P < 0.05$) lower median reduction (1.32 log unit) compared to that from equipment surfaces (1.48 log unit). A similar trend was observed for *E. coli* populations from carcasses that had not been (1.38 log unit) or had been (1.49 log unit) exposed to antimicrobial interventions, i.e., after hide-on and wash air chilling ($P < 0.05$).

**Phylogenetic distribution of *E. coli* as characterized by serotyping, source of isolation, and lactic acid resistance.** To gain insight into the relationships between the genotypic and phenotypical characteristics in relation to acid resistance and phylogeny of *E. coli*, a core-gene phylogenetic tree was constructed with the 178 genomes (Fig. 2). The distribution of seven *E. coli* populations from different processing stages, acid phenotype, genes related to acid resistance, and commonly identified serotypes across phylogenetic groups are shown in Table 1. Associations of *E. coli* populations, serotypes, gene prevalence with the phylogeny, or source of isolation were also determined (Table 2).

The 178 *E. coli* isolates belonged to phylogenetic groups A, B1, B2, C, D, and E. Diversification in acid phenotypes was across phylogroups (Fig. 2). Groups A and B1 were the most commonly identified phylogroups in the seven *E. coli* populations, together accounting for 93% of the total (Table 1). The exception was for the *E. coli* population from

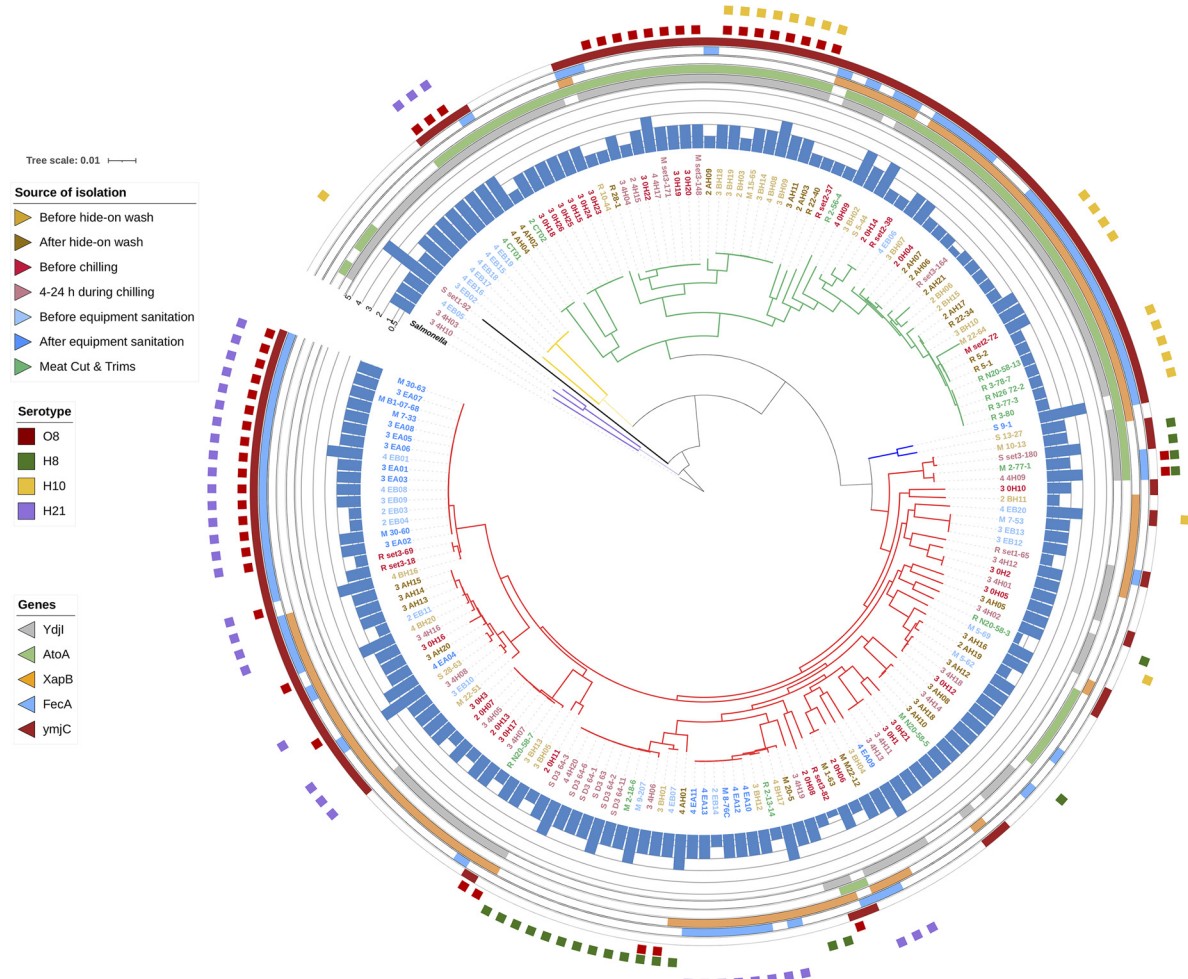

**FIG 2** Phylogeny of *E. coli* from beef processing environments. A core phylogenetic tree was constructed using 178 *E. coli* genomes. Branches were color-coded to represent different phylogroups: A (green), B1 (red), B2 (black), C (blue), D (purple), and E (yellow). Isolates were color-labeled according to stages of processing or characteristics shown in the legend on the left. A bar chart around the tree indicates the degree of acid resistance of each isolate, with the heights of the bars being proportional to the log reduction by lactic acid treatment. In the next layers, the color strips indicate the presence of genes overrepresented in the acid-resistant *E. coli*. In the outermost layer, symbols with different colors represent the four major serogroups.

equipment surface after sanitation, which was 95% of phylogroup B1. In addition, the *E. coli* population from equipment surface before sanitation was the most diverse group, consisting of all five phylogroups identified in this study. *E. coli* from different processing stages did not differ in the proportion of *E. coli* of phylogroup B1 (*P* > 0.05), but the proportion of phylogroup A differed significantly (Table 1). Of the *E. coli* populations from carcasses, the proportion of phylogroup A was similar across different populations and composed approximately 50% of each population, except for the *E. coli* population collected during carcass chilling, accounting for 19.4% of the population.

There were 55 O groups and 32 H types predicted for the 178 *E. coli* isolates (Table 3). Of these, O8, H21, H8, and H10 were the most abundant, accounting for 68% of the total isolates (Table 3). These O and H groups were randomly distributed among the phylogroups and *E. coli* from different processing stages, except for *E. coli* isolated after equipment sanitation (Fig. 2). Interestingly, most of *E. coli* (17/19) recovered after equipment sanitation shared either O8, H21, or the combination of O8:H21. In addition, *E. coli* O8:H21 and O8:H10 were clustered together in groups B1/E and A, respectively.

Of the 20 genes overrepresented in acid-resistant *E. coli*, 8 were part of the *ydj* operon coding for a novel carbohydrate pathway, six belonged to the *ato* operon required for

**TABLE 1** Proportions of phylogroups and serogroups of generic *E. coli* as affected by location and stage of processing at the slaughter plant

| Stages of processing | Phylogroups (%) | | | | | | Serogroups (%) | | | |
|---|---|---|---|---|---|---|---|---|---|---|
| | A[a] | B1 | B2 | C | D | E | O8 | H21 | H10 | H8 |
| Hide-on wash | | | | | | | | | | |
| Before | 55.56 | 40.74 | 0.00 | 3.70 | 0.00 | 0.00 | 40.74 | 11.11 | 33.33 | 3.70 |
| After | 46.67 | 53.33 | 0.00 | 0.00 | 0.00 | 0.00 | 10.00 | 26.67 | 10.00 | 16.67 |
| Chilling | | | | | | | | | | |
| Before | 45.45 | 54.55 | 0.00 | 0.00 | 0.00 | 0.00 | 15.15 | 3.03 | 0.00 | 0.00 |
| During 4 to 24 h | 19.35 | 70.97 | 3.23 | 0 | 6.45 | 0 | 19.35 | 3.23 | 6.45 | 35.48 |
| Equipment sanitation | | | | | | | | | | |
| Before | 4.17 | 66.67 | 0 | 0 | 4.17 | 25.00 | 25.00 | 19.17 | 4.17 | 8.33 |
| After[a] | 0 | 94.74 | 0 | 5.26 | 0 | 0 | 63.16 | 84.21 | 0.00 | 0.00 |
| Fabrication | 57.14 | 42.86 | 0 | 0 | 0 | 0 | 14.29 | 14.29 | 35.71 | 14.29 |

[a]$P < 0.05$.

short-chain fatty acid degradation, three were part of the *fec* operon coding for a $Fe^{2+/3+}$ transportation system, two were part of *xapABR*, which codes for the xanthosine catabolism pathway, and one (*ymjC*) code for a putative oxidoreductase (Table 4). The other components of the *fec* and *xap* operons had a prevalence between 60 and 64%, which were below the arbitrary 65% threshold for the acid-resistant *E. coli* (Table S2 in the supplemental materials.). The distribution of *xap* and *fec* genes varied in *E. coli* associated with different phylogenetic groups and processing stages (Table S3). The prevalence of all *ydj* and *ato* genes differed significantly among the phylogenetic groups and *E. coli* from different processing stages (Table S3); thus, *ydjI* and *atoA* coding for the key enzymes in their respective operons were selected to illustrate the distribution of the operons (Fig. 2). The abundance of the *ydj* or *ato* operons was lower by >20% in *E. coli* populations from equipment surfaces compared to other *E. coli* populations (Table 2). For *E. coli* after equipment sanitation, the prevalence of *ydj* or *ato* operons was the lowest among all populations of *E. coli*; in contrast, the prevalence of *fecAIR* was the highest among the *E. coli* populations. Such distinct distribution was not observed for *xapBR* and *ymjC*. The prevalence of *ydj* or *ato* genes in phylogroup A and B1 differed by >50%, whereas less variation was noted for *xapBR* and *fecAIR*. *E. coli* positive for these genes were more resistant to the lactic acid treatment than *E. coli* without them ($P < 0.05$; Fig. S2).

**Genome diversity of *E. coli* recovered from different sources.** To elucidate the effects of beef processing on the genome diversity of *E. coli*, pairwise average nucleotide identity (ANI) values of *E. coli* genomes from the same processing stages were calculated

**TABLE 2** Prevalence of genes over-represented in acid-resistant *E. coli* as affected by stage of processing and within phylogroup[a]

| Genes | Prevalence (%) in *E. coli* | | | | | | | | Phylogroups | | | | | | |
|---|---|---|---|---|---|---|---|---|---|---|---|---|---|---|---|
| | Hide-on wash | | Chilling | | Equipment sanitation | | | | | | | | | | |
| | Before | After | Before | During 4 to 24 h | Before | After | Fabrication | P value < 0.05 | A | B1 | B2 | C | D | E | P value < 0.05 |
| *ydjI* | 70.4 | 60 | 72.7 | 58.1 | 33.3 | 5.3 | 78.6 | * | 96.6 | 29.9 | 100 | 50 | 100 | 100 | ** |
| *atoA* | 59.3 | 56.7 | 54.5 | 35.5 | 12.5 | 5.3 | 64.3 | * | 98.3 | 11.2 | 100 | 100 | 66.7 | 0 | * |
| *xapB* | 59.3 | 56.7 | 48.5 | 22.6 | 37.5 | 36.8 | 57.1 | | 50.8 | 44.9 | 0 | 100 | 0 | 0 | * |
| *xapR* | 63 | 56.7 | 48.5 | 25.8 | 37.5 | 36.8 | 57.1 | | 52.5 | 45.8 | 0 | 100 | 0 | 0 | * |
| *fecA* | 37 | 36.7 | 33.3 | 12.9 | 33.3 | 89.5 | 57.1 | * | 47.5 | 37.4 | 0 | 50 | 0 | 0 | |
| *fecI* | 37 | 36.7 | 36.4 | 22.6 | 33.3 | 89.5 | 57.1 | * | 54.2 | 37.4 | 0 | 50 | 0 | 0 | * |
| *fecR* | 37 | 36.7 | 36.4 | 16.1 | 33.3 | 89.5 | 57.1 | * | 52.5 | 37.4 | 0 | 50 | 0 | 0 | * |
| *ymjC* | 70.4 | 70 | 54.5 | 29 | 41.7 | 68.4 | 64.3 | * | 89.8 | 41.1 | 100 | 50 | 0 | 0 | * |

[a]Differences of gene prevalence among different processing stages or phylogroups were indicated by a single or double asterisk(s), respectively.

**TABLE 3** Summary of serogroups[a] of the *E. coli* isolates as determined by *in silico* serotyping

| Typing (total no.) | Subtypes | No. (%) of genomes |
|---|---|---|
| O (55) | O8 | 45 (25) |
| | O9 | 11 (6) |
| | O89, O149, O154 | 5−10 (<6) |
| H (32) | H21 | 38 (21) |
| | H8 | 21 (11) |
| | H10 | 20 (11) |
| | H7, H9, H11, H12, H16, H2, H20, H25, H39 | 5−10 (<6) |

[a]O groups or H types with less than 2% prevalence are not shown.

to estimate the genome similarity (Fig. 3). Higher ANI indicates a lower genomic diversity. *E. coli* populations recovered after chilling and equipment sanitation were less diverse than those recovered before the same processing stages, as indicated by median ANI. However, *E. coli* recovered after chilling had the widest range of ANI values among the *E. coli* populations. Of the seven populations, *E. coli* isolated after sanitation had the least genome diversity. The genome relatedness of the *E. coli* population on cuts/trim was higher than that of *E. coli* from carcasses but lower than *E. coli* from equipment after sanitation.

**Genome similarity of *E. coli* sharing the same serogroup or serotype.** The O groups and H types were dispersed across the phylogeny; however, some serotypes clustered in lineages, i.e., O8:H21 and O8:H10 (Fig. 2). To provide a quantitative measure for the phylogenetic relatedness of strains sharing the same serotypes, *E. coli* of the same O group, H type, or serotype were compared in genome similarity estimated by ANI (Fig. 4). Generally, *E. coli* sharing either the same O group or H type had a higher genome diversity than *E. coli* sharing both O and H types (serotype) ($P < 5 \times 10^{-6}$); however, an exception was found for *E. coli* O8:H8. The genome relatedness of *E. coli* O8:H8 was greater than *E. coli* O8 but did not differ from *E. coli* H8. *E. coli* strains with serotypes O8:H10 and O8:H21 were among the most closely genomically related, and their genome similarity was higher than that of other O8, H10, or H21 *E. coli* ($P < 5 \times 10^{-6}$). In addition, *E. coli* O8:H21 or O8:H10 had the median ANI of 100%, reflecting the close phylogenetic relation shown in Fig. 2. The lack

**TABLE 4** Genes overrepresented in acid-resistant *E. coli*

| Gene cluster | Gene | Function |
|---|---|---|
| Ydj | *ydjE* | MFS transporter |
| | *ydjF* | Putative deoR/GlpR family DNA-binding transcription regulator |
| | *ydjG* | Putative aldo/keto reductase or NADH-dependent methylglyoxal reductase |
| | *ydjH* | Carbohydrate kinase family protein or kinase, PfkB family |
| | *ydjI* | Ketose-bisphosphate aldolase |
| | *ydjJ* | Putative zinc-dependent dehydrogenase |
| | *ydjK* | Putative MFS transporter |
| | *ydjL* | Putative zinc-dependent dehydrogenase |
| Xap | *xapB* | Xanthosine permease |
| | *xapR* | Transcriptional activator |
| Fec | *fecA* | $Fe^{3+}$ outer membrane porin |
| | *fecI* | RNA polymerase, sigma factor |
| | *fecR* | Regulator, periplasmic |
| Ato | *atoA* | Acetoacetyl-CoA transferase, beta subunit |
| | *atoB* | Acetyl-CoA acetyltransferase |
| | *atoD* | Acetoacetyl-CoA transferase, alpha subunit |
| | *atoC* | Regulatory protein |
| | *atoE* | Putative short-chain fatty acid transporter |
| | *atoS* | Signal transduction histidine-protein kinase |
| NA[a] | *ymjC* | Putative oxidoreductase |

[a]NA, not applicable.

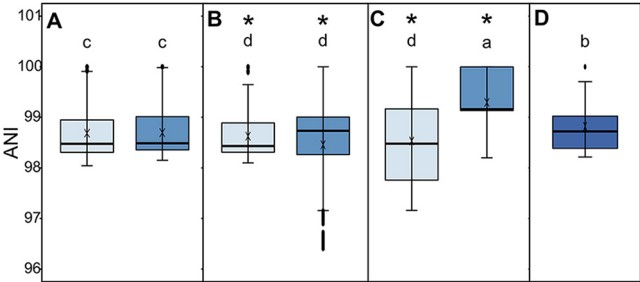

**FIG 3** Boxplots of average nucleotide identity (ANI) for genomes of *E. coli* isolates recovered from different processing stages in beef plants. *E. coli* was recovered from carcasses at hide-on wash (A), carcasses at chilling (B), equipment surfaces at sanitation (C), and fabrication (D). In panels A to C, before (light blue) and after (dark blue) a processing stage are color coded. The median ANI is indicated by a line; the mean ANI is indicated by ×; outliers are indicated by filled dots. A single asterisk indicates significant differences of ANI between groups in the same panel; different letters indicate significant differences among groups across panels A to D ($P < 0.05$).

of serotype diversity in the equipment isolates may indicate persisting strains in the meat processing environment.

## DISCUSSION

*E. coli* on meat might be derived from multiple ecological niches and have been exposed to various environmental stressors at different processing stages, resulting in changes in the population diversity and stress resistance. Individual strains of *E. coli* may not always be appropriate for assessing the impact of meat processing on the response of *E. coli* populations to biocides/interventions in commercial circumstances, because of the genome diversity of *E. coli* (25, 26). A large data set of *E. coli* at a population level is thus representative of the ecology of *E. coli* in the meat plant environment and for quantifying response to stress under conditions relevant to meat plants. However, studies of *E. coli* in the meat plant environment have primarily been characterized by phenotyping and genotyping using subtyping methods (22, 27, 28). The information at the genomic level, especially for generic *E. coli*, was previously scarce. Therefore, the present study determined the resistance to lactic acid of a large population of *E. coli* from various stages of processing in beef packing plants and their relation to the phylogeny and genome diversity to better understand the role of environmental stressors on the population structure and resistance to lactic acid of *E. coli* in the meat plant environment. Furthermore, the comparative genomic analysis of 178 *E. coli* revealed differences in genes associated with acid phenotypes and phylogeny between animal- (carcass preintervention) and equipment-associated *E. coli*.

**Effect of beef processing practices on the resistance to lactic acid.** The efficacy of the lactic acid treatment depends on its concentration and is also affected by other

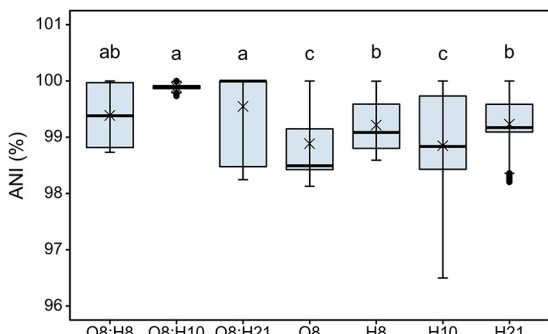

**FIG 4** Boxplots of average nucleotide identity (ANI) for genomes of *E. coli* isolates of the most common serogroups/-types identified in this study. Genomes sharing O8, H8, H10, and H21, including serotypes O8: H21, O8:H8, and O8:H10 H21, accounted for >68% of the total genomes analyzed. Boxes represent the range of ANI of genomes sharing the same serotypes/groups. The median ANI is indicated by a line; the mean ANI is indicated by ×; outliers are indicated by the black dots. Different letters indicate significant differences in ANI of genomes among different subtypes ($P < 5 \times 10^{-6}$).

factors in commercial plants, e.g., the operation condition and bacterial physiological state (11). Solutions of lactic acid at 2% initially introduced to decontaminate beef carcasses have been ineffective against *E. coli* in commercial plants (29, 30). The concentration was subsequently increased to 4 to 5%, which achieves an approximately >2 log reduction of *E. coli* and up to 3.3 log reduction of aerobes (31, 32). A previous study has assessed the relative response of generic *E. coli* from beef carcasses/trim under conditions similar to the present study, i.e., 5% lactic acid at pH 2.9 for 1 h resulted in a median reduction of 0.9 log (22), which is close to the median reduction of 1.07 log for *E. coli* isolated from carcasses before chilling in this study. Of note, the incubation temperatures used in the previous and the present studies differed slightly, 4°C versus 0°C (ice water). Despite the log reduction for individual strains ranging from <0.5 to >5 log unit, the median reduction varied from 1.55 log unit (after hide-on wash) to 1.07 (before chilling) for the seven *E. coli* populations (this study). There was no increase in acid resistance noted when comparing *E. coli* collected from meat products to that collected from the hides of carcasses, or comparing *E. coli* populations after and before a processing step, including *E. coli* populations from carcasses or equipment surfaces exposed and not exposed to biocides. Even so, significant differences were observed between populations of *E. coli* associated with animal and equipment surfaces, which also exhibited a different prevalence of genes associated with the resistance to lactic acid. These findings suggest that beef packing practices under which the samples were collected did not lead to an increased level of resistance to lactic acid in generic *E. coli*, but resulted in the diversification of individual *E. coli* strains in respect to lactic acid resistance.

**Relation of genetic makeup to lactic acid resistance of *E. coli*.** Acid stress induces cellular damage in multiple ways, such as high turgor pressure impairing the stability and activities of enzymes, which disrupts the electron transport chain for cellular repair, leading to the accumulation of reactive oxygen species (ROS) (23). Both organic and inorganic acids (lactic acid, acetic acid, or HCl) induce the expression of universal acid response genes, including those involved in the oxidative, envelope, and cold shock stress responses, and iron and manganese uptake; however, acidulant-specific genes were also found (24). It has been demonstrated that, in response to exposure to lactic acid, *E. coli* increased the expression of genes involved in ROS response, TCA cycle, and energy metabolism, including the metabolism of propionate, glyoxylate, arginine, and lysine, and degradation of fatty acid, which contributed to proton exportation (24, 33, 34). Some of these lactic-acid induced stress responses were common to other acid treatments (24). However, this does not fully explain the mode of action of acid resistance marking the difference in acid phenotypes. A transcriptomic study demonstrated that strain-specific genes involved in oxidative stress response, DNA damage repair, and iron uptake resulted in differences in acid resistance between *E. coli* K-12 and *E. coli* O157:H7 (24). Similar results were found in this study by genome comparison using a large number of *E. coli* with differing resistance to lactic acid.

The present study found that genes involved in short-chain fatty acid (SCFA) degradation (*ato* operon) and energy metabolism (*ydj* operon, *xap* genes) were more abundant in acid-resistant than other *E. coli*. The *ato* system was required for aerobic growth of *E. coli* on SCFAs (35) and was also involved in acetoacetate or acetate degradation (36). SCFAs are products of the gut microbiome, and they induce acid stress by passing the cell membrane in their undissociated form, thereby reducing intracellular pH (37). The *ydi* system coding for a novel carbohydrate pathway was characterized in two studies (38, 39). However, the enzymatic functions of the *ydj* operon were only partially determined (38, 39). YdjH and YdjI preferentially catalyzed the long-chain monosaccharides with carboxylate terminals (38, 39). The best substrate of YdjI was *L-glycero-L-galacto-octuluronate*; however, it has no natural abundance. A similar compound, *d-glycero-d-talo-octuluronate*, was found in the lipopolysaccharide of acid-resistant *Acetobacter pasteurianus* (40). The exact role of *ato* and *ydj* in acid response remains to be determined.

In addition to the two full operons *ato* and *ydj*, the *xap* and *fec* operons were partially overrepresented in the acid-resistant *E. coli*. The differences in prevalence among

*xap* or *fec* genes could have arisen from various factors such as loss/gain of genes or limitations of draft genomes (41). The *xap* genes involved in the xanthosine metabolism allow *E. coli* to utilize purine nucleoside as a carbon source (42). The *fec* operon consists of *fec*ABCDE, *fecI*, and *fecR*, which are involved in the iron dicitrate transport system, contributing to iron uptake (43, 44). Accumulation of iron is well documented under ROS stress (45). The *fecI* and *fecR* are the regulatory components of the *fec* operon, both of which were upregulated upon the formation of hydroxyl radicals (46). The *fecI* was upregulated in *E. coli* exposed to organic and inorganic acid (24). Overall, the specialization of these genes in acid-resistant *E. coli* could contribute to acid resistance and carbon source utilization.

**Relations of genetic makeup, phylogeny, and ecology of *E. coli*.** Although host associations are poorly correlated with the phylogeny of *E. coli*, phylogroup A and B1 were frequently represented by the vertebrate commensals (1, 47), including bovine (48 to 50). To survive in the gastrointestinal environment of animals and humans, *E. coli* is required to survive the low pH in the stomach and SCFAs in the intestine (23). The higher abundance of *ato* and *ydi* operons in phylogroup A and preintervention carcass-associated *E. coli* largely from animals suggests their enzymatic functions may provide ecological benefits to *E. coli* through utilizing novel carbohydrate substrates and SCFAs, and combating stress in the environment rich in SCFAs, such as ruminant digestive systems. The lack of such conditions in environments such as meat processing may result in the otherwise lower prevalence of genes in equipment-associated *E. coli* populations.

In contrast with the higher prevalence of the *ydj* and *ato* operons in the preintervention carcass-associated *E. coli*, *fec* genes were more abundant in equipment-associated *E. coli*. Biocides used for equipment sanitation, such as chlorine-based or quaternary ammonium compounds (QACs), are effective for killing *E. coli* at concentrations much lower than the in-use concentrations (51). However, in meat processing facilities, there could be niches where the concentrations of biocides could be at sublethal levels, for instance, underneath food debris and in biofilms. Sublethal concentrations of chlorine and QACs induce oxidative stress (52, 53), likely contributing to the high prevalence of *fec* genes in equipment-associated *E. coli*.

The population structure of meat plant *E. coli* populations shifted from phylogroups A and B1 at relatively equal proportions for the preintervention carcass-associated populations (before hide-on wash and before chilling) to being dominated by phylogroup B1 (before sanitation) or almost exclusively phylogroup B1 (after sanitation) for the equipment-associated *E. coli* populations. A number of studies have indicated the shifted population structure of *E. coli* from different sources toward phylogroup B1 as affected by environments. A study by Touchon et al. in which over 1,000 commensal *E. coli* isolates from various habitats were analyzed in relation to phylogeny and population structure has found that the water isolates had the highest fraction of phylogroup B1, compared to the *E. coli* populations from humans, nonhuman mammals, and birds (26). When different phylogroups of *E. coli* from humans and cattle were tested for survival in water, *E. coli* strains of phylogroup B1 seemed to be the persistent group in water even though the utilization of different carbon sources or macromolecules did not differ between phylogroups or between persistent and control groups (54). Additionally, *E. coli* from crops is mainly of group B1 and has a different pattern of carbon source utilization from host-adapted *E. coli* (55). Taken together, phylogroup B1 *E. coli* exhibited a stronger adaptation to the beef plant environment, i.e., nonhuman and animal reservoirs. Strong environmental adaptation and recurrent transition between environmental niches (56) likely lead to the abundance of phylogroup B1 in the environment. On the other hand, *E. coli* has a high genome plasticity/diversity, with <10% of the pan-genome being core genes (25) and distinct accessory genomes in phylogroups revealed by large-scale genomic analyses (26, 41).

In conclusion, the beef processing environment did not select for a lactic acid-resistant population but led to variations in serotypes, phylogenetic distribution, and acid resistance in *E. coli*. In beef packing plants, the routes of contamination often include multiple media,

**TABLE 5** *Escherichia coli* isolates included in this study

| Beef plants ID | Isolation source | Process | Processing stages | No. of isolates for acid resistance analysis | No. of genomes for genomic analysis | Reference |
|---|---|---|---|---|---|---|
| B | Hide-on carcasses | Hide wash | Before | 100 | 27 | 13 |
| | Hide-off carcasses | | After | 100 | 30 | |
| A | Dressed carcasses | Carcass chilling | Before | 100 | 33 | 16, 59 |
| | | | During 4 to 24 h | 100 | 31 | |
| A | Fabrication equipment surfaces | Equipment sanitation | Before | 100 | 24 | 12 |
| | | | After | 100 | 19 | |
| A | Beef cuts and trim | | | 100 | 14 | 60 |

which could not trace back to a single source, e.g., animals (28). Equipment surface was considered to be a source of recurring contamination of *E. coli* on meat products, although cattle are significant initial contributors of *E. coli* (57, 58). Thus, it is debatable to continuously use *E. coli* as an indicator for fecal contamination without considering the environment/process under evaluation, because of its broad range of niches, as reflected in the changes in the population structure as it flows through the beef processing stages. Environmental factors may then enrich a subpopulation of *E. coli* originating from animals. However, it is challenging to trace the immediate origin of *E. coli*, which has been naturalized to the secondary habitats, as it could not be differentiated by phenotyping methods. Subtyping or genotyping methods based on metabolic genes, such as *ydj* and *ato* genes that are associated with ecological function, have the potential to predict the primary hosts or environmental niches of *E. coli*. Further functional analysis of the Ydj pathway would shed light on the role of the novel carbohydrate pathway in the evolution history of *E. coli*.

## MATERIALS AND METHODS

**Bacterial strains and culture conditions.** A total of 700 *E. coli* isolates were included in this study, representing populations from different processing stages. They were originally isolated from two federally inspected beef plants (A and B) in Alberta, Canada from 2013 to 2014 (Table 5) (12, 13, 16, 59, 60). In brief, 200 isolates were recovered from plant B, including 100 each from carcasses before and after a hide-on carcass wash. Another 500 *E. coli* isolates were recovered from plant A, including 100 each from carcasses before and after carcass chilling, from equipment surfaces before and after sanitation, and from beef cuts and trim. At plant A, carcasses were trimmed and washed with cold water, followed by air chilling for up to 67 h. No antimicrobial interventions were otherwise applied to carcasses or meat products at this plant at the time of sample collection. The daily sanitation process at Plant A at the time of sample collection included rinsing with pressurized water (40 to 50°C), cleaning with chlorine-based alkaline foaming agents, and sanitizing with a QAC sanitizer (12). Plant B differed from Plant A in that multiple antimicrobial interventions were applied to carcasses. These included hide-on carcass wash with 1.5% sodium hydroxide (55°C), spraying skinned carcasses with 5% lactic acid, and pasteurization of carcass sides with steam (>90°C) at the time of sample collection (13, 17). *E. coli* isolates from carcasses before hide-on wash at Plant B and before air chilling at Plant A were regarded as preintervention carcass-associated, as no antimicrobial interventions had been applied to carcasses at these stages. *E. coli* recovered from carcasses after air chilling and after hide-on wash were classified as isolates exposed to antimicrobial interventions. Equipment-associated *E. coli* isolates are those recovered from equipment surfaces before and after equipment sanitation.

Each *E. coli* glycerol stock culture archived at −80°C was streaked onto MacConkey agar (Oxoid Canada Inc., Mississauga, ON, CA) and subsequently incubated at 35°C for 24 h. A single colony was inoculated into Luria-Bertani broth (LB; BD Difco, Fisher Scientific, Canada), followed by incubation for 16 h at 35°C and 80 rpm. The identity of each isolate was verified by PCR using primers targeting *uidA* (5′-TGTTACGTCCTGTAGAAAGCCC-3′ and 5′-AAAACTGCCTGGCACAGCAATT-3′) (61).

**Acid treatment.** The overnight culture of *E. coli* was 10-fold diluted in LB broth and was used for acid treatment. Acid treatment was carried out in LB containing 5.5% (vol/vol) of 90% lactic acid (Sigma-Aldrich, Oakville, ON, CA) with the pH adjusted to around 2.9, followed by sterilization using 0.02-$\mu$m filtration units (Nalgene, VWR, Edmonton, AB, CA). Aliquots of 100 $\mu$L of cell suspension in LB broth of each strain were added to 0.9 mL acidified LB broth and LB broth without the addition of lactic acid or adjustment of pH (control), followed by incubation in ice water for 1 h. At the end of the treatment, 100 $\mu$L of treated and control samples were transferred into 9.9 mL of phosphate-buffered saline solution (PBS) (Fisher Scientific, Ottawa, ON, Canada) to neutralize the solution and stop the treatment. After neutralization, the pH of the cell suspension in PBS ranged from 6.5 to 6.8 as measured by a pH meter

(Fisherbrand accumet AP115, Fisher Scientific). Cell suspensions in PBS solution were serial-diluted in PBS, and then 1 mL of appropriate dilutions was plated onto Aerobic 3M Petrifilm Aerobic Count Plates (3M Corp., St. Paul, MN, USA). Plates were incubated at 35°C for 18 h. The colonies were then counted following the manufacturer's instructions and regarded as surviving *E. coli*.

**Whole-genome sequencing, assembly, and annotation.** The isolates for sequencing were selected to include different acid phenotypes and represent populations from different processing stages. *E. coli* strains with reductions of cell counts <1, 1.5 to 3.5, or >5 $\log_{10}$ CFU/mL were regarded as acid resistant, mediocre, and sensitive. All *E. coli* having log reductions of <1 ($n = 45$) or >5 ($n = 1$) were selected. Due to the larger number of *E. coli* with cell reductions of 1.5 to 3.5 $\log_{10}$ CFU/mL, *E. coli* of this group were further selected by source of isolation to generate a population with even distribution across different processing stages. This selection scheme resulted in a total of 178 isolates. Of the 178 isolates, genomes of 122 isolates had been obtained in previous studies (19 from BioProject PRJNA716667, 103 from BioProject PRJNA819951; Table S1). The remaining 56 *E. coli* isolates were whole-genome sequenced in this study.

DNA was isolated from overnight cultures of *E. coli* grown in 5 mL of LB broth, using a Qiagen DNeasy blood and tissue kit (Qiagen, Toronto, ON, Canada) according to the manufacturer's instructions. DNA samples were prepared into shotgun libraries and sequenced using an Illumina NovaSeq 6000 platform (150 bp, paired-end [PE]) with an estimated sequencing depth >300× (Genome Quebec, QC, CA). The quality of reads was assessed using FastQC v0.11.7, and the adaptor sequences and low-quality reads were removed by Trimmomatic 0.39, using default parameters (62). Assemblies were obtained by SPAdes v3.14.0 with the most optimal k-mer value for each genome (63). The quality of the assemblies was assessed by Quast v5.0.2 (64). Genomes were annotated using Prokka v1.14.6 (https://github.com/tseemann/prokka).

***In silico* sero- and phlyotyping.** The serotype of each genome was predicted, and the species of *E. coli* were confirmed using ECTyper v1.0 (https://github.com/phac-nml/ecoli_serotyping) with default settings. Then, the phylogroups were determined by Clermont Typing (https://github.com/A-BN/ClermonTyping), an *in silico* PCR method using primer sequences designed previously (65).

**Average nucleotide identity (ANI).** The antigens O8, H21, H8, and H10 were among the most commonly occurring O- and H- serogroups of *E. coli* examined in this study (Table S1). For these serogroups or serotypes, isolates sharing the same O groups and/or H types or collected from different processing stages were pairwise compared for ANI (%) using FastANI 1.32 (66).

**Core genome phylogenetic analysis.** Core genome alignments of the 178 *E. coli* were used to construct a core-genome phylogenetic tree. The pan-genome of the isolates was parsed using Roary v3.13.0 (67). Genes shared by >99% of the genomes with >95% nucleotide identity were regarded as core genes ($n = 2,601$). The maximum likelihood phylogenetic tree based on the alignment of core genes was constructed using RAxML with the general time reversible gamma nucleotide model (GTRGAMMA) and bootstrapping for 1,000 replicates (68). *Salmonella enterica* subsp. *enterica* serovar Typhimurium str. LT2 (GenBank assembly accession: GCA_000006945.2) was included as an outgroup (69).

**Detection of genes associated with acid resistance.** Of the 178 isolates sequenced, 45 had log reductions <1 from the lactic acid treatment, and 1 strain had a reduction >5. The remaining 132 isolates had reductions between 1 and 5 log units. To screen for genes potentially contributing to the survival of *E. coli* of lactic acid treatment, a genome-wide association between two groups was conducted: the 45 acid-resistant isolates that had log reduction of <1, and the others (133 isolates), which had log reductions ≥1. First, genes that were differentially present in the two groups were screened by Scoary v 1.6.16 (https://github.com/AdmiralenOla/Scoary), with the following criteria: $P < 0.05$ by Fisher's exact test, and present in >65% acid-resistant *E. coli* but absent in >50% of the rest of *E. coli* ($n = 92$). Then, the genes obtained through the above analyses were confirmed using Abricate 1.0.1 (https://github.com/tseemann/abricate) (Table S2). Briefly, the nucleotide sequences of the 92 genes from Scoary analysis were retrieved from the Roary pan-genome analysis output and made into a local database. All *E. coli* sequences (178 genomes) were searched against the local database using Abricate with >95% identity and 80% coverage. Significant differences in the gene distribution were tested by Fisher's exact test ($P < 0.05$). Three genes related to hypothetical functions, or prophages (one prophage repressor gene), or with redundant functions were excluded from the downstream analysis. After removing these genes, 20 genes remained, which were considered to be associated with acid resistance and were further analyzed.

**Statistical analysis.** Cell counts were log-transformed, and the reduction of cell counts was calculated as $\log_{10}(N_0/N)$, with $N_0$ and $N$ representing cell counts of *E. coli* before (control) and after the lactic acid treatment, respectively. Data of log reductions and ANI did not follow a normal distribution, as determined by the Shapiro-Wilk test ($P < 0.05$). Therefore, significant differences among the median log reductions and the median ANI of *E. coli* between different groups were compared using the Wilcoxon test for pairwise comparison or the Kruskal-Wallis test and Dunn's test for multipairwise comparison. Fisher's exact test was used to determine the differences in the distribution of genes in different phylogenetic groups in *E. coli* from different processing stages, and the association of *E. coli* from different processing stages with phylogeny and serotypes. Significant differences in each analysis were determined by a $P$ value of <0.05.

**Data availability.** Genome data are available under BioProject PRJNA816492.

## SUPPLEMENTAL MATERIAL

Supplemental material is available online only.
**SUPPLEMENTAL FILE 1**, XLSX file, 2.5 MB.
**SUPPLEMENTAL FILE 2**, PDF file, 0.2 MB.

## ACKNOWLEDGMENTS

We acknowledge the funding support by the Beef Cattle Research Council (FOS01.17) in Canada for this study. We also thank Peipei Zhang, Hui Wang, Arun Kommadath, and Frances Tran for their technical support and bioinformatics analysis expertise.

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
