## [Reviewer comments · Microbiology Spectrum]

Microbiology Spectrum

Lactic Acid Resistance and Population Structure of Escherichia coli from Meat Processing Environment

Yuan Fang, Kim Stanford, and Xianqin Yang

Corresponding Author(s): Xianqin Yang, Agriculture and Agriculture-Food Canada

Review Timeline:

Submission Date:	April 14, 2022
Editorial Decision:	August 5, 2022
Revision Received:	September 5, 2022
Accepted:	September 9, 2022

Editor: Luca Cocolin

Reviewer(s): Disclosure of reviewer identity is with reference to reviewer comments included in decision letter(s). The following individuals involved in review of your submission have agreed to reveal their identity: Alex Gill (Reviewer #1)

Transaction Report:

DOI: <https://doi.org/10.1128/spectrum.01352-22>

August 5, 2022

Dr. Xianqin Yang
Agriculture and Agriculture-Food Canada
6000 C&E Trail
Lacombe
Canada

Re: Spectrum01352-22 (Acid Resistance and Population Structure of *Escherichia coli* from Meat Processing Plants Environments)

Dear Dr. Xianqin Yang:

Link Not Available

Sincerely,

Luca Cocolin

Journals Department
Reviewer comments:

Reviewer #1 (Comments for the Author):

A well designed study approprietly reported and analysed. There are a small number of points in the text where the meaning is unclear or the sentence structure is clumsy clear. These are indicated in annotations to a copy of the manuscript file (uploaded with this review)

Reviewer #2 (Public repository details (Required)):

Deposition numbers from this study and two previous (isolates are included in this study) are provided. I am unable to access two (BioProject PRJNA819951 and BioProject PRJNA816492) of the three using the numbers provided. This should be checked.

Reviewer #2 (Comments for the Author):

The study is generally technical sound and the data is useful in particular because a relatively large collection of strains are used. Some issues should be addressed:

1. Even with the description and summary provided in Table 1 it is difficult to understand the selection of isolates. I needed to read this a few times before I understood it. The authors should clarify this section (perhaps a diagram would be useful instead of a Table).
2. The authors acidify using lactic acid only. However, in the title and throughout they extrapolate this to acidification in general. While I appreciate the isolates were previously exposed to lactic acid it is not at all apparent (as the authors will know from the literature) that the strains would behave the same when exposed to a different organic acid or inorganic acid. In order to reflect the findings the authors must refer specifically to lactic acid in the title and throughout where appropriate. They should also include some discussion on the role different acids may play in the Discussion section.

Staff Comments:

Preparing Revision Guidelines

Please return the manuscript within 60 days; if you cannot complete the modification within this time period, please contact me. If you do not wish to modify the manuscript and prefer to submit it to another journal, please notify me of your decision immediately so that the manuscript may be formally withdrawn from consideration by Microbiology Spectrum.

**Acid Resistance and Population Structure of *Escherichia coli* from Meat Processing Plants Environments**

Yuan Fang¹, Kim Stanford², and Xianqin Yang^{1*}

¹Agriculture and Agri-Food Canada Lacombe Research and Development Centre, 6000 C & E Trail, Lacombe, Alberta T4L
1W1, Canada

²University of Lethbridge, Lethbridge, Alberta, Canada

**Running title:** Response of meat plant *E. coli* to lactic acid treatment

* Author for correspondence

Xianqin Yang

Tel: 1-403-782-8119

email: xianqin.yang@agr.gc.ca

**Abstract**

To explore the effect of beef processing on ~~the genotype and phenotype of~~ *Escherichia coli* populations in relation to acid resistance,
~~the this~~ study investigated the links among acid response, phylogenetic structure, genome diversity, and genotypes associated with acid
resistance of meat plant *E. coli*. Generic *E. coli* isolates (n=700) were from carcasses, fabrication equipment, and beef products. Acid
treatment was carried out in Luria-Bertani broth containing 5.5% lactic acid (pH 2.9). Log reductions of *E. coli* ranged from <0.5 to >5
log CFU/mL (median:1.37 log). No ~~increase difference~~ in acid resistance was observed ~~for-between~~ *E. coli* populations recovered before
and after a processing step or antimicrobial interventions. ~~*E. coli* from The~~ pre-intervention carcasses ~~associated *E. coli*~~ were slightly
more resistant than ~~*E. coli* isolated from~~ equipment-~~associated *E. coli*~~, differing by <0.5 log unit. Acid-resistant *E. coli* (log reduction
<1, n=45) had a higher prevalence of genes related to energy metabolism (*ydj*, *xap*, *ato*) and oxidative stress (*fec*, *ymjC*) than the ~~other~~
~~population~~ less resistant *E. coli* (log reduction >1, n=133). The *ydj* and *ato* operons were abundant in pre-intervention carcass-associated
*E. coli*. In contrast, *fec* genes were abundant in ~~equipment surface associated *E. coli*~~ from equipment surfaces. ~~The pre-intervention *E.*~~
~~*coli* populations~~ contained ~~shifted from~~ phylogroups A and B1 in relatively equal proportions. ~~for the pre-intervention carcass associated~~
~~populations to being predominated by or primarily of p~~Phylogroup B1 predominated (95%) ~~for-in~~ the population from equipment-
~~associated populations~~. Of note, *E. coli* collected after sanitation shared either the antigens of O8_h or H21. Additionally, genome diversity
decreased after chilling and equipment sanitation. Overall, beef processing did not select for acid-resistant *E. coli* but shaped the
population structure.

**Importance**

Antimicrobial interventions have significantly reduced the microbial loads on carcasses/meat products; however, the wide use of
chemical and physical biocides has raised concerns over their potential for selecting resistant populations in the beef processing
environment. Phenotyping of acid resistance and whole-genome analysis described in this study demonstrated beef processing practices
led to ~~variations-differences~~ in acid resistance, genotype, and population structure between carcass- and equipment-associated *E. coli*
but did not select for the acid-resistant population. Results indicate that genes coding for the metabolism of long-chain sugar acids (*ydj*)
and short-chain fatty acids (*ato*) were more prevalent in carcass-associated than equipment-associated *E. coli*. These results suggest *E.*
*coli* from carcasses and equipment surfaces have been ~~driven-by~~exposed to different ~~evolution-forees~~selective pressures. The findings
~~better-improve~~ our understanding of the microbial ecology of *E. coli* in food processing environments and in general.

**Introduction**

*Escherichia coli* is an inhabitant of the intestines of warm blood animals (1), but some strains are pathogenic to humans and animals
(2). Cattle are well-recognized ~~reservoirs-hosts~~ of Shiga toxin-producing *E. coli* (STEC), ~~including e.g.,~~ *E. coli* O157:H7, making beef
products susceptible to STEC contamination (3, 4). Fecal shedding of *E. coli* results in a heavy microbial load on animal hides, ~~more so~~
~~for particularly~~ animals raised in centralized feedlot environments (5-7). The transmission of *E. coli* to beef carcasses is regarded as
inevitable during the process of dehiding because the opening cuts are from the outer to the inner surface of the hides and ~~the~~ aerosols
are generated by the motion of hide pulling (7, 8). Concerns over STEC and other microbiological hazards ~~in general and risks from~~
~~STEC~~ led to the requirement in the United States of America for testing of carcasses for *E. coli*, and beef trimmings for *E. coli* O157:H7
and six non-O157 STEC (9). The prevalence and level of pathogenic *E. coli* on carcasses are much lower than ~~those of~~ generic *E. coli*,
which could render population-based studies impossible without sampling an extremely high number of samples. Generic *E. coli* is thus
an appropriate indicator to assess the hygiene performance of meat processing (10).

To address ~~these~~ concerns regarding the microbiological safety of meat ~~safety~~, various physical and chemical antimicrobial interventions
are implemented at beef plants in North America (7, 11, 12). Commonly used antimicrobial interventions in large beef processing
facilities include hide-on carcass wash with 1.5% sodium hydroxide at 55 °C (13, 14), carcass wash with organics acids, such as 4-5%
lactic acid and peroxyacetic acid at various concentrations, and pasteurization of carcasses with steam or hot water at 85 °C or >90 °C
(15). Air-chilling can effectively reduce the microbial load on carcasses by up to 2 log units (16) and can also be regarded as an
intervention step. *E. coli* on most chilled carcasses can be reduced to <1 CFU/10,000 cm² when the microbial contamination of meat

Commented [R1]: Reference is to US regulation only. US and Canadian and EU regulation and guidance to control STEC O157 in meat predates 2012.

<https://inspection.canada.ca/preventive-controls/meat/raw-beef-products/eng/1541538060346/1541538137261>

during carcass dressing is well controlled, and decontamination for carcasses is effective (17). Consequently, the decreased number of
foodborne illnesses caused by *E. coli* O157:H7 in Canada has been attributed to the implementation of antimicrobial interventions at
beef processing plants (18).

Lactic acid at 4-5% is one of the most effective chemical interventions for reducing the microbial load of carcasses, including *E. coli* in
commercial beef processing settings (7, 17). However, complete elimination of bacteria on carcasses by any antimicrobial interventions
is not yet attainable, largely due to the structure and/or composition of carcass surfaces and raw meat (13, 17). In addition to the
application of antimicrobial interventions to decontaminate carcasses/meat products, various biocides, such as caustic chlorine cleaners,
quaternary compounds and peroxyacetic acid based-sanitizers are routinely used to clean and sanitize equipment to minimize
recontamination of meat during meat fabrication (12, 19). Accordingly, the wide use of physical/chemical agents in meat processing
facilities may select for bacteria with elevated resistance to antimicrobial interventions.

*E. coli* is generally acid-tolerant (20-22). To survive in acidic environments, *E. coli* utilizes various ~~acid resistance~~ physiological
mechanisms coupled with the buffering capacity of cytoplasmic macromolecules, such as proteins, amino acids, and polyphosphate, and
the reduction of proton influx mediated by membrane fluidity and cytoplasmic chaperones, as well as consumption of intracellular
protons through decarboxylases systems (23). Additionally, secondary carbohydrate metabolism and ATP-dependent metabolic
pathways have also been suggested to be involved in supporting *E. coli* ~~to survive~~ survival under acidic conditions (24). The ability to
balance intracellular pH and to recover from the cellular damages caused by low pH enhances the survival of *E. coli* in acid conditions.

Commented [R2]: Many resistance mechanisms protect against a variety of stresses, not just acid.

If the current meat packing practices with respect to antimicrobial interventions and usage of biocides in sanitation lead to increased
acid resistance in *E. coli* in the beef processing environments, it would not only compromise the efficacy of lactic acid as an antimicrobial
intervention but may also contribute to the ~~development-increased prevalence~~ of ~~a-acid~~ resistant *E. coli* ~~population~~ in the broader
environment. Therefore, the present study aimed to 1) determine the acid response of 700 *E. coli* collected from different stages of beef
processing to evaluate whether the beef processing environment selects for acid-resistant *E. coli*, and 2) explore the relationships among
the acid phenotypes, phylogeny, source of isolation, and genetic makeup of *E. coli* to unravel the mechanisms driving differences in
acid resistance.

**Material and Methods**

**Bacterial strains and culture conditions.** A total of 700 *E. coli* isolates were selected from the culture collection of the Lacombe
Research and Development Centre of Agriculture and Agri-Food Canada and these *E. coli* were originally isolated from two federally-
inspected beef plants (A and B) in Alberta from 2013-2014 (**Table 1**) (12, 13, 16, 25, 26). At plant A, carcasses were trimmed and
washed with cold water, followed by air-chilling for up to 67 h. No antimicrobial interventions were otherwise applied to carcasses or
meat products at this plant at the time of sample collection. The daily sanitation process at Plant A at the time of sample collection
included rinsing with pressurized water (40-50°C), cleaning with chlorine-based alkaline foaming agents, and sanitizing with a
quaternary ammonium compound (QAC) sanitizer (12). ~~Plant B differed Different~~ from Plant A ~~in that~~, multiple antimicrobial
interventions were applied to carcasses. ~~These at Plant B, which~~ included hide-on carcass wash with 1.5% sodium hydroxide (55 °C),
spraying skinned carcasses with 5% lactic acid, and pasteurization of carcass sides with hot steam (>90°C) at the time of sample

collection (13, 17). In brief, 100 isolates each from carcasses before and after hide-on wash at Plant B, and from carcasses before and
during air-chilling, equipment before and after sanitation, and meat products at Plant A were included in this study. *E. coli* isolates from
carcasses before hide-on wash at Plant B and before air-chilling at Plant A were regarded as pre-intervention carcass-associated, as no
antimicrobial interventions had been applied to carcasses at these stages. *E. coli* recovered from carcasses after air-chilling and after
hide-on wash were classified as isolates exposed to antimicrobial interventions. Equipment-associated *E. coli* isolates ~~were referred to~~
as are those recovered from equipment surfaces before and after equipment sanitation.

Each *E. coli* glycerol stock culture archived at -80°C was streaked onto MacConkey agar (Oxoid Canada Inc., Mississauga, ON, CA)
and subsequently incubated at 35 °C for 24 h. A single colony was inoculated into Luria-Bertani broth (LB; BD Difco, Fisher Scientific,
Canada), followed by incubation for 16 h at 35°C and 80 rpm. The identity of each isolate was verified by PCR using primers targeting
*uidA* (5'-TGTTACGTCCTGTAGAAAGCCC-3', and, 5'-AAAACCTGCCTGGCACAGCAATT-3') (27).

**Acid treatment.** The overnight culture of *E. coli* was 10-fold diluted in LB broth and was used for acid treatment. Acid treatment was
carried out in LB containing 5.5% (v/v) of 90% lactic acid (Sigma-Aldrich, Oakville, ON, CA) with the pH adjusted to around 2.9,
followed by sterilization using 0.02 µm filtration units (Nalgene®, VWR, Edmonton, AB, CA). Aliquots of 100 µL of cell suspension
in LB broth of each strain were added to 0.9 mL acidified LB broth and LB broth without the addition of lactic acid or adjustment of
pH (control), followed by incubation in ice water for 1 hour. At the end of the treatment, 100 µL of treated and control samples were
transferred into 9.9 mL of phosphate-buffered saline solution (PBS) (Fisher Scientific, Ottawa, ON, Canada) to neutralize the solution

and stop the treatment. After neutralization, the pH of the cell suspension in PBS ranged from 6.5 to 6.8 as measured by a pH meter
(Fisherbrand™accuMET™ AP115, Fisher Scientific). Cell suspensions in PBS solution were serial-diluted in PBS, and then 1 mL of
appropriate dilutions was plated onto Aerobic 3M Petrifilm™ Aerobic Count Plates (3M Corp., St. Paul, MN, USA). Plates were
incubated at 35 °C for 18 h. The colonies were then counted following the manufacturer's instructions and regarded as surviving *E. coli*.

**Whole-genome sequencing, assembly and annotation.** All *E. coli* strains with reductions of cell counts < 1 or $> 5 \log_{10}$ CFU/mL were
selected. Isolates with cell reductions between 1.5-3.5 \log_{10} CFU/mL were randomly selected but which would result in a roughly even
distribution across different stages of processing. A total of 178 isolates among the 700 *E. coli* used for acid treatment were included.
Of the 178 isolates, genomes of 122 isolates had been obtained in previous studies (18 from BioProject PRJNA716667, 104 from
BioProject PRJNA819951; **Table S1**). The remaining 56 *E. coli* isolates selected based on their acid phenotype and isolation
sources/processing stages were whole-genome sequenced in this study.

DNA was isolated from overnight cultures of *E. coli* grown in 5 mL of LB broth, using a QIAGEN DNeasy Blood & Tissue kit
(QIAGEN, Toronto, ON, Canada) according to the manufacturer's instructions. DNA samples were prepared into shotgun libraries and
sequenced using an Illumina NovaSeq 6000 platform (150 bp, PE) with an estimated sequencing depth $> 300 \times$ (Genome Quebec, QC,
CA). The quality of reads was assessed using FastQC v0.11.7, and the adaptor sequences and low-quality reads were removed by
Trimmomatic 0.39, using default parameters (28). Assemblies were obtained by SPAdes v3.14.0 with the most optimal k-mer value for

Commented [R3]: Meaning unclear. Where the isolates selected for genome analysis, randomly without reference to the isolation source? Or was the selection weighted to ensure representation from different sources? How was random selection conducted? Was it truly random (i.e. strains selected by random number)?

[revised manuscript text omitted]
 ~~indicative of in~~ commercial circumstances due to the genome diversity ~~ification~~ of *E. coli* (36, 37). A large dataset
of *E. coli* at a population level is thus more ~~pertinent to represent~~ representative of the ecology of *E. coli* in ~~the~~ meat plant environments
and for quantifying the stress response to stress under conditions relevant to meat plants. However, studies of *E. coli* in the meat plant
environments have primarily been characterized by phenotyping and genotyping using subtyping methods (22, 38, 39). The information
at the genomic level, especially for generic *E. coli*, was previously scarce. Therefore, the present study determined the acid resistance
of a large population of *E. coli* from various stages of processing in beef packing plants and their relation to the phylogeny and genome
diversity of *E. coli* to better understand the role of environmental stressors on the population structure and acid resistance of *E. coli* in

Commented [R4]: How many strains in each group. If n is low this observation would seem less significant.

Commented [R5]: Could this be evidence of persistent strains in the processing environment?

[revised manuscript text omitted]

- 63. Tezel U, Pavlostathis SG. 2015. Quaternary ammonium disinfectants: microbial adaptation, degradation and ecology. *Current Opinion in*
*Biotechnology* 33:296-304.
- 64. Wang Z, Fang Y, Zhi S, Simpson DJ, Gill A, McMullen LM, Neumann NF, Gänzle MG. 2020. The locus of heat resistance confers resistance
to chlorine and other oxidizing chemicals in *Escherichia coli*. *Appl Environ Microbiol* 86:e02123-19.

- 65. Berthe T, Ratajczak M, Clermont O, Denamur E, Petit F. 2013. Evidence for coexistence of distinct *Escherichia coli* populations in various
aquatic environments and their survival in estuary water. *Appl Environ Microbiol* 79:4684-93.
- 66. Méric G, Kemsley EK, Falush D, Siggers EJ, Lucchini S. 2013. Phylogenetic distribution of traits associated with plant colonization in
*Escherichia coli*. *Appl Environ Microbiol* 15:487-501.
- 67. Ishii S, Sadowsky MJ. 2008. *Escherichia coli* in the environment: implications for water quality and human health. *Microbes and*
*Environments* 23:101-108.
- 68. Wang R, Kalchayanand N, King DA, Luedtke BE, Bosilevac JM, Arthur TM. 2014. Biofilm formation and sanitizer resistance of *Escherichia*
*coli* O157:H7 strains isolated from “high event period” meat contamination. *Journal of Food Protection* 77:1982-1987.
- 69. Sinde E, Carballo J. 2000. Attachment of *Salmonella* spp. and *Listeria monocytogenes* to stainless steel, rubber and
polytetrafluorethylene: the influence of free energy and the effect of commercial sanitizers. *Food Microbiology* 17:439-447.
- 70. Yang X, Badoni M, Tran F, Gill CO. 2015. Microbiological effects of a routine treatment for decontaminating hide-on carcasses at a large
beef packing plant. *J Food Prot* 78:256-63.
- 71. Yang X, Tran F, Youssef MK, Gill CO. 2015. Determination of sources of *Escherichia coli* on beef by multiple-locus variable-number
tandem repeat analysis. *J Food Prot* 78:1296-302.

**Table 1 *Escherichia coli* isolates included in this study.**

Beef plants ID	Isolation source	Process	Processing stages	No. of isolates for acid resistance analysis	No. of genomes for genomic analysis	Reference
B	Hide-on carcasses	Hide wash	Before	100	27	(70)
	Hide-off carcasses		After	100	30	
A	Dressed carcasses	Carcass chilling	Before	100	33	(16, 25)
			During 4-24 h	100	31	
A	Fabrication equipment surfaces	Equipment sanitation	Before	100	24	(12)
			After	100	19	
A	Beef cuts and trim	-		100	14	(71)

466

467

468 **Table 2 Summary of serogroups^a of the *E. coli* isolates as determined by *in silico***
 469 **serotyping.**

Typing (total no.)	Subtypes	No. (%) of genomes
O (55)	O8	45 (25)
	O9	11 (6)
	O89, O149, O154	5-10 (<6)
H (32)	H21	38 (21)
	H8	21 (11)
	H10	20 (11)
	H7, H9, H11, H12, H16, H2, H20, H25, H39	5-10 (<6)

470 ^aO groups or H types with less than 2% prevalence were not shown.

471

472 **Table 3. Genes overrepresented in acid-resistant *E. coli*.**

Gene Cluster	Gene	Function
Ydj	ydjE	MFS transporter
	ydjF	Putative deoR/GlpR family DNA-binding transcription regulator
	ydjG	Putative aldo/keto reductase or NADH-dependent methylglyoxal reductase
	ydjH	Carbohydrate kinase family protein or kinase, PfkB family
	ydjI	Ketose-bisphosphate aldolase
	ydjJ	Putative zinc-dependent dehydrogenase
	ydjK	Putative MFS transporter
	ydjL	Putative zinc-dependent dehydrogenase
Xap	xapB	Xanthosine permease
	xapR	Transcriptional activator
Fec	fecA	Fe ³⁺ outer membrane porin
	fecI	RNA polymerase, sigma factor
	fecR	Regulator, periplasmic
Ato	atoA	Acetoacetyl-CoA transferase, beta subunit
	atoB	Acetyl-CoA acetyltransferase
	atoD	Acetoacetyl-CoA transferase, alpha subunit
	atoC	Regulatory protein
	atoE	Putative short-chain fatty acid transporter
	atoS	Signal transduction histidine-protein kinase
NA	ymjC	Putative oxidoreductase

473 NA: not applicable

474

Response to reviewers' comments

The authors would like to thank the reviewers for their time and effort reviewing the manuscript, and for their constructive comments for improving the work. Detailed response to each comment is below.

Reviewer #1 (Comments for the Author):

A well designed study appropriately reported and analysed. There are a small number of points in the text where the meaning is unclear or the sentence structure is clumsy. These are indicated in annotations to a copy of the manuscript file (uploaded with this review).

Re: Thanks for the meticulous editing and comments. We have revised the manuscript according to reviewer's edits. Reviewer's comments were addressed as below:

R3: Meaning unclear. Where the isolates selected for genome analysis, randomly without reference to the isolation source? Or was the selection weighted to ensure representation from different sources? How was random selection conducted?: Was it truly random (i.e. strains selected by random number)?

Re: Additional information on the selection procedure is included in the section of material and methods under "Whole-genome sequencing, assembly and annotation".

R4: How many strains in each group. If n is low this observation would seem less significant): That's a good point.

*Re: The statistical significance was assessed carefully with the $P < 5 \times 10^{-6}$ to address the biases caused by sample size. The sample size of pairwise ANI analysis is 2^n (n =no. of isolates), which would give a relatively large number for statistical analysis. The detailed information of serogroup was included in S1. The total number of *E. coli* that shares the same H or O was shown in Table 2.*

R5: Could this be evidence of persistent strains in the processing environment?

Re: That's a good point. A sentence has been added to reflect this point.

Reviewer #2 (Public repository details (Required)):

Deposition numbers from this study and two previous (isolates are included in this study) are provided. I am unable to access two (BioProject PRJNA819951 and BioProject PRJNA816492) of the three using the numbers provided. This should be checked.

Re: The data has been deposited and is set to be released upon publication of this manuscript.

Reviewer #2 (Comments for the Author):

The study is generally technically sound and the data is useful in particular because a relatively large collection of strains are used. Some issues should be addressed:

1. Even with the description and summary provided in Table 1 it is difficult to understand the

selection of isolates. I needed to read this a few times before I understood it. The authors should clarify this section (perhaps a diagram would be useful instead of a Table).

Re: Thanks for reviewer's suggestion. The description for table 1 and selection for the isolates were re-written (Bacterial strains and culture condition) for better clarify.

2. The authors acidify using lactic acid only. However, in the title and throughout they extrapolate this to acidification in general. While I appreciate the isolates were previously exposed to lactic acid it is not at all apparent (as the authors will know from the literature) that the strains would behave the same when exposed to a different organic acid or inorganic acid. In order to reflect the findings the authors must refer specifically to lactic acid in the title and throughout where appropriate. They should also include some discussion on the role different acids may play in the Discussion section.

Re: Thanks for the comments. Agree with reviewer's suggestions. We have updated the MS as suggested, including 1) specifying acid to lactic acid in the tile, subtitles and places where are needed; 2) Adding a discussion about different acids on acid stress responses under the "Relation of genetic makeup to acid resistance of E. coli".

September 9, 2022

Dr. Xianqin Yang
Agriculture and Agriculture-Food Canada
6000 C&E Trail
Lacombe
Canada

Re: Spectrum01352-22R1 (Lactic Acid Resistance and Population Structure of Escherichia coli from Meat Processing Environment)

Dear Dr. Xianqin Yang:

Your manuscript has been accepted, and I am forwarding it to the ASM Journals Department for publication. You will be notified when your proofs are ready to be viewed.

Sincerely,

Luca Cocolin
Editor, Microbiology Spectrum

Journals Department
Table S1-3: Accept

Fig S1-2: Accept